# Effects of Early Weaning on Performance and Carcass Quality of Nellore Young Bulls

**DOI:** 10.3390/ani14050779

**Published:** 2024-03-01

**Authors:** Gabriela Abitante, Paulo Roberto Leme, Matheus Sousa de Paula Carlis, Germán Darío Ramírez-Zamudio, Bianca Izabelly Pereira Gomes, Luiza Budasz de Andrade, Rodrigo Silva Goulart, Guilherme Pugliesi, Arlindo Saran Netto, Carl Robertson Dahlen, Saulo Luz Silva

**Affiliations:** 1Department of Animal Science, Faculdade de Zootecnia e Engenharia de Alimentos, Universidade de Sao Paulo, Pirassununga 13635-900, SP, Brazil; gabriela.abitante@usp.br (G.A.); prleme@usp.br (P.R.L.); dicarlis@usp.br (M.S.d.P.C.); germanramvz@usp.br (G.D.R.-Z.); biancaizabelly@usp.br (B.I.P.G.); budasz@usp.br (L.B.d.A.); rogoulart@usp.br (R.S.G.); saranetto@usp.br (A.S.N.); 2Department of Animal Reproduction, School of Veterinary Medicine and Animal Science, Universidade de Sao Paulo, Pirassununga 13635-900, SP, Brazil; gpugliesi@usp.br; 3Department of Animal Science, North Dakota State University, Fargo, ND 58108, USA; carl.dahlen@ndsu.edu

**Keywords:** animal nutrition, beef cattle, beef production, feeding system, meat quality

## Abstract

**Simple Summary:**

Early weaning is a practice that holds the potential to enhance beef cattle production systems. It aims to improve the reproductive performance of cows in the subsequent breeding season while minimizing the impact on weaned calves when they are appropriately nutritionally managed. This study aimed to compare the effects of early weaning (weaning at 150 days; EW) and conventional weaning (weaning at 240 days; CW) on the growth, carcass characteristics, and meat quality of Nellore bulls. A total of 74 non-castrated calves were assigned to an EW (*n* = 37) or CW (*n* = 37) strategy. During approximately 400 days in the growth phase, both groups received supplementation during grazing, followed by a finishing period in a feedlot for 87 days. The results showed that calves in the EW group were lighter than the CW group at the time of conventional weaning, yet this weight difference did not impact their growth or meat quality during the fattening phase. This study suggests that early weaning, combined with adequate supplementation, can be a viable and sustainable approach in beef cattle production without compromising the animals’ productive development or meat quality.

**Abstract:**

This study compared early weaning (EW; 150 days) with conventional weaning (CW; 240 days) in Nellore young bulls, evaluating performance, carcass characteristics, and meat quality. A total of 74 non-castrated male calves were divided into two weaning strategies: EW (*n* = 37) and CW (*n* = 37). During the growth phase, which lasted 454 ± 14 d for EW calves and 359 ± 16 d for CW calves, animals received a protein-energy supplement at a ratio of 5 g per kg of body weight while grazing Brachiaria brizantha cv. Marandu. The animals were managed for an 87d finishing phase in three collective feedlot pens, with a 3-week adaptation protocol, starting with corn silage to a concentrate ratio of 55:45 and reaching a ratio of 30:70 in the final diet. Body weight, average daily gain (ADG), dry matter intake (DMI), feed efficiency (FE), carcass characteristics, and meat quality were evaluated. The EW group was approximately 44 kg lighter than the CW at the time of conventional weaning (*p* < 0.001). However, this weight difference did not influence ADG, DMI, and FE in the finishing phase. No significant differences were observed in carcass characteristics such as yield percentage, loin area, subcutaneous fat thickness, and meat quality, except for the weight of primal cuts, which was greater in the CW group (*p* < 0.001). Thus, although calves weaned early are lighter throughout subsequent production phases than those weaned conventionally, performance, efficiency, carcass yield, and meat quality are not affected.

## 1. Introduction

In the past decade, beef production in Brazil has experienced an evolution due to the intensification of the feedlot industry [1]. However, over ca. 80% of the beef produced in Brazil comes from pasture-finished cattle [2]. Furthermore, the breeding herd, predominantly comprised of zebu cows, is managed in extensive forage-based systems, with calves weaned between 7 and 8 months of age [3]. However, in the Nellore breed, milk production does not meet the calves’ nutritional requirements after the third month of lactation [4], and prolonged lactation can also interfere with the reproductive performance of the dam during the subsequent breeding season [5].

Early weaning is an effective practice to improve the reproductive performance of dams, as it allows the energy requirements of lactation to be directed towards achieving mature body weight in primiparous cows [6], replenishing the necessary body reserves to sustain the high demands of late gestation [7], and ensure pregnancy in the following breeding season [8,9]. Regarding calves, early weaning alone can compromise post-weaning growth performance, such as average daily gain and body weight [10,11]. However, some studies suggest that when early weaning is associated with adequate nutrition, post-weaning performance can be similar to calves weaning later [12,13]. 

Early weaning is not a commonly adopted practice in Brazilian beef cattle farming. Although this practice is not aimed at surpassing the performance of traditionally weaned calves, the combination of early weaning with supplementation strategies can result in significant improvements in post-weaning growth, as well as positively impacting carcass and meat quality [14,15]. According to a recent study by Pedro, et al. [16], adopting weaning at 120 days of age in Nellore calves and a supplementation regimen demonstrated similar growth compared to those weaned at 205 days without supplementation. Moreover, in this study, early weaning associated with a supplementation regimen positively affected genes with the potential to increase intramuscular fat deposition in meat.

In this study, we hypothesize that calves weaned early, despite being lighter, are not affected in performance, carcass characteristics, and meat quality compared to traditionally weaned calves. Thus, the objective was to evaluate the performance, carcass characteristics, and meat quality of Nellore young bulls subjected to early weaning (150 days of age) or traditional weaning (240 days of age).

## 2. Materials and Methods

The experiment was conducted at the College of Animal Science and Food Engineering (FZEA) of the University of São Paulo (USP), located in the municipality of Pirassununga, São Paulo, Brazil, under the approval of the Commission on Ethics in Animal Use, protocol number CEUA: 2884250620.

### 2.1. Animals, Treatments, Management, and Diet

A total of 74 bull calves were used in two treatments: Early weaning (EW; 148 ± 8 days) or conventional weaning (CW; 242 ± 10 days). Calves were born between August and October 2020 and were divided into four groups at the beginning of the experiment based on the date of birth and the cow’s category (primiparous or multiparous). The groups were formed to ensure that the calves had similar ages and were used as a criterion for the division into the EW and CW treatments. A schematic of experimental treatments and the timeline is presented in Figure 1.

During the pre-weaning phase, calves were kept with their respective dams in *Brachiaria brizantha* cv. *Marandu* pastures. At an average age of 90 days, all calves began receiving protein-energy supplementation via creep-feeding, offering 0.5% of body weight (BW) per bull per day (Table 1). The supplement was administered three times a week until weaning, which occurred at either 150 or 240 days. Each group underwent two weaning procedures, totaling eight weanings, with the same procedure for all groups. Weaning procedures included weighing all calves in the group, regardless of the treatment (EW or CW). Calves were subjected to a health protocol at weaning including deworming with Ivermectin 1% (Ivermectin^®^, Vansil, Descalvado, SP, Brazil) and Levamisole (Ripercol L^®^ 150F, Zoetis, Campinas, SP, Brazil), and vaccination against clostridiosis (Poli-Star^®^, MSD Saúde Animal, Cruzeiro, SP, Brazil) following the manufacturer’s recommendations. 

After weaning, the calves were moved to another area and managed under a rotational grazing system as a single group. Pasture rotation occurred weekly, based on grass (*Brachiaria brizantha* cv. *Marandu*) height. Fresh forage samples were manually collected by hand-plucking along the pen to assess the nutritional composition during both the dry season (April–September) and the rainy season (October-March). The average nutritional composition during the rainy season was 18.5% dry matter, 9.5% crude protein, 1.8% ether extract, 39.1% acid detergent fiber, and 69.6% neutral detergent fiber. In contrast, during the dry season, the composition was 79.3% dry matter, 3.7% crude protein, 0.8% ether extract, 40.6% acid detergent fiber, and 76.3% neutral detergent fiber. All calves received protein-energy supplementation equivalent to 0.5% of their BW daily throughout the growing phase, which lasted 454 ± 14 days for the EW group and 359 ± 16 days for the CW group. Additionally, during this period, they were administered an inactivated rabies vaccine (Labovet, Feira de Santana, BA, Brazil) and a keratoconjunctivitis preventive vaccine (Bioqueratogen^®^ Óleo, Biogénesis Bagó, Buenos Aires, BA, Argentina).

At 20 ± 0.4 months of age and a mean BW (±SD) of 458.4 ± 37.1 kg, 70 bulls were transferred to the experimental feedlot. A total of 4 bulls were removed from the trial due to problems unrelated to the treatments (3 from the EW group and 1 from the CW group). Bulls were divided into three blocks based on BW and housed in three collective pens equipped with Intergado^®^ electronic feeders and waterers (Betim, MG, Brazil). 

Prior to the animals entering the feedlot, a health protocol was implemented involving deworming using Doramectin 1% (Exceller^®^, MSD, Cruzeiro, SP, Brazil) and Albendazole Sulfoxide 15% (Agenbendazol^®^, Agener União Saúde Animal, São Paulo, SP, Brazil), along with a multivalent vaccination (Starvac^®^, Labovet, Feira de Santana, BA, Brazil). Additionally, the bulls were identified with RFID electronic ear tags and weighed after a 14-h fasting period.

The first 28 days of the feedlot period were dedicated to adapting the bulls to the facilities and the diet. During the initial 7 days, corn silage and mineral salt were offered ad libitum, while in the following 21 days, the concentrate content was gradually increased using the step-up method with a 7-day interval until reaching the final diet (Table 1). After the completion of the adaptation phase, 66 bulls remained (*n* = 31 for EW and *n* = 35 for CW; 3 bulls did not adapt to the facilities and were removed from the trial). At the end of the feeding period, the distribution of bulls by treatment in each block was as follows: in block 1, there were 10 bulls from the EW group and 12 from the CW group; in block 2, both the EW and CW groups contained 11 bulls; and in block 3, there were 10 EW bulls and 12 CW bulls.

### 2.2. Performance and Ultrasound Measurements

During the growing phase, calves were weighed approximately every 30 days without fasting. The BWs were used to adjust the supplement supply during the growing phase. In the finishing phase, bulls underwent a 14-h fasting period before initial and final weighing. The average daily weight gain (ADG) was calculated for the finishing phase using the formula ADG, kg/day = final BW − initial BW/Finishing period days.

During the finishing phase, the daily feed intake on an as-fed basis was recorded for each bull through the Intergado^®^ system (Betim, MG, Brazil). Based on the system data, dry matter intake (DMI) was calculated by adjusting as fed intake to diet dry matter (DM) content. Silage samples were dried at 105 °C for 24 h once a week to adjust the DM content of the diet. The DM analyses of the concentrate ingredients were conducted each time a new batch of concentrate was prepared. Feed efficiency (FE) during the finishing period was calculated by dividing ADG by DMI for each bull.

Carcass ultrasound measurements were collected at 0, 33 and 61 days on feeding, using Exago ultrasound equipment (IMV Technologies Co, l’Aigle, Orne, France), equipped with a 180 mm 3.5 MHz linear array transducer. The ribeye area (REAu), SFTu between the 12th and 13th thoracic vertebrae, and rump fat thickness (RFTu) corresponding to the *Gluteus medius* muscles were taken [17]. The obtained images were analyzed by an experienced technician using Lince^®^ V.1.5 software (M&S Consultoria Agropecuária Ltd., Pirassununga, SP, Brazil).

After an average of 87 days on feed (ranging from 74 to 97 days), bulls were selected for slaughter based on the BW criteria and ultrasound subcutaneous fat thickness (SFTu) above 3 mm. Bulls were harvested in six groups of 11 animals over an 18-day period.

### 2.3. Slaughter and Carcass Evaluation

The bulls were harvested in six groups (11 animals per group) within 18 days, at the slaughterhouse of USP located approximately 300 m from the feedlot facilities. For harvesting, animals were selected based on treatments (EW and CW), BW and SFTu (minimum of 3 mm), prioritizing those with the highest body weight and similar subcutaneous fat thickness.

Bulls were stunned using a captive bolt and then bled from the jugular vein, following the humane slaughter practices stipulated in the Regulations for Sanitary Inspection and Industrialization of Animal Origin Products [18]. 

After removing the skin, head, feet, and viscera, the hot carcass weight (HCW) was recorded. The dressing percentage (DP) was calculated as follows: DP, % = HCW/final BW × 100. Subsequently, the weights of the kidney, pelvic, and inguinal fat (KPIF) were recorded at the time of slaughter. The weights of the reticulum-rumen, omasum, abomasum, and intestines were recorded after evisceration and removal of their contents. After 24 h of chilling (4 °C), the carcasses were weighed to obtain the chilled carcass weight (CCW). The chilling loss was calculated using the following equation: Chilling loss, % = [(HCW − CCW)/HCW] × 100. Subsequently, the left half-carcass of each bull was subdivided into hindquarter, forequarter, and combined plate, flank, and short ribs. The hindquarter was deboned to determine the weights and percentages of retail cuts, bones, and trim.

The pH and temperature of the *Longissimus thoracis* (LT) muscle at the 12th rib level were measured 1 and 24 h after slaughter using a digital thermometer (Incoterm, Porto Alegre, RS, Brazil) and a pH meter equipped with a penetration probe (model HI8314, Hanna Instruments, São Paulo, SP, Brazil).

### 2.4. Meat Quality Analyses

Twenty-four hours after chilling, three steaks (2.54 cm thick) were collected from the LT muscle between the 10th and 12th thoracic vertebrae of the left half of the carcass. Immediately after cutting, two steaks were individually vacuum-packed and refrigerated at 2 °C for 7 and 14 days for subsequent analysis of color, cooking loss (CL), and Warner–Bratzler shear force (WBSF). Meanwhile, the third steak from each bull was analyzed immediately.

To measure color, samples were removed from the vacuum packages, allowed to bloom for 30 min and then L*, a*, and b* values were estimated on the surface of the steaks, at three different points, using a CM2500d spectrophotometric colorimeter (Konica Minolta Brazil, São Paulo, SP, Brazil), with a 10 mm aperture size, illuminant A, and 10° observer angle [19].

Cooking loss and WBSF were evaluated following the methodology outlined by the American Meat Science Association [20]. Initially, the steaks were weighed to determine their initial weight. Then, a thermometer was inserted into the geometric center of each sample, which was subsequently heated in an industrial electric oven (model F130/L, Fornos Elétricos Flecha de Ouro Ind. e Com. Ltd.a., São Paulo, SP, Brazil) at 170 °C until the internal temperature of the samples reached 40 °C. After reaching the desired temperature, samples were flipped and allowed to continue cooking in the oven until the internal temperature reached 71 °C. Subsequently, samples were removed from the oven, cooled to room temperature (22 °C), and weighed. The cooking loss value was obtained as a percentage using the following formula: CL, % = ((Initial weight − final weight)/initial weight) × 100.

After weighing, the steaks were wrapped in plastic film and refrigerated at 4 to 6 °C for 12 h. To evaluate WBSF, six cores with a diameter of 1.27 cm, parallel to the orientation of the muscle fibers, were collected using a bench drill. These cores were subjected to a shearing test at a 200 mm/min rate using a Warner–Bratzler blade attached to a TMS-PRO texture analyzer (Food Technology Corporation, Sterling, VA, USA). The WBSF was determined as the average of the values obtained from the six cylindrical samples, and the results were expressed in Newtons (N).

### 2.5. Statistical Analysis

The data were analyzed using the MIXED procedure of SAS (SAS, version 9.4, Inc., Cary, NC, USA). A randomized block design based on the initial weight of the animals was adopted for analyzing the weights during the growing and finishing phases, ultrasound measurements, and meat quality at various post-mortem times. 

Repeated measures over time, such as body weight, carcass ultrasonography, and analyses of aged meat, the weaning treatment (EW at 150 days vs. CW at 240 days), evaluation time (body weight in growing and finishing phases; carcass ultrasonography on days 1, 34, and 62; and meat analyses on days 1, 7, and 14 post-mortem) and treatment × time interaction were considered as fixed effects. Meanwhile, the weaning group (based on the four birth groups), cow category (primiparous or multiparous), block, and slaughter group were used as random effects. Covariance matrices were tested, and the one that showed the best fit was selected. For characteristics with a single measurement (feedlot performance, carcass traits after slaughter), the weaning treatment was considered a fixed effect and the block as a random effect.

Effects were considered significant when *p* values were less than 0.05 and indicated a trend when 0.05 < *p* ≤ 0.10.

## 3. Results

### 3.1. Performance in the Growing and Finishing Phases and Carcass Ultrasound

There was a significant treatment × age interaction BW during the growing phase (*p* < 0.001; Figure 2). The BW was not different at the time of EW, but CW calves were heavier (*p* ≤ 0.001) by d 240 and at every subsequent evaluation during the growing period. 

The BW of the CW group was higher (*p* < 0.001) than the EW throughout the finishing period; at feedlot entry, CW bulls were ~36 kg heavier than EW (Table 2). At the end of the finishing period, the BW difference between the groups remained, with the CW bulls being ~41 kg heavier than the EW bulls (*p* < 0.001). Additionally, CW bulls tended to have a greater feed intake than EW bulls during the feedlot period (*p* = 0.083). However, no treatment effect on ADG and FE was observed.

Regarding carcass ultrasound measurements, no interaction was observed between the weaning group and day of evaluation (Figure 3). The REAu, SFTu, and RFTu values increased with time on feed (*p* < 0.001). The REAu was greater for CW bulls (*p* = 0.002); in contrast, the RFTu was greater for the EW group (*p* = 0.014). No effect was observed for the SFTu (*p* = 0.605), except for the day of evaluation (*p* < 0.001).

### 3.2. Carcass Characteristics and Retail Cuts Yield

The CW bulls had heavier carcasses compared to EW bulls (*p* < 0.001; Table 3), but no difference was found for dressing percentage, REA, STF, or kg of KPIF. However, KPIF expressed as a percentage of carcass weight was greater for EW compared to CW (*p* = 0.008).

Bulls from the CW group had heavier primal cuts (forequarter, thin flank, and hindquarter) than EW bulls (*p* < 0.001), but no difference was observed when primal cuts were expressed as the percentage of carcass weight (Table 3). In addition, CW bulls had heavier (*p* < 0.001) retail cuts and bone in the hindquarter than EW bulls, but the kg of trim was similar between treatments. Though retail cuts and bone as a percentage of hindquarter weight were not impacted by treatment, EW bulls had a greater (*p* = 0.055) percentage of trim in the hindquarter compared to CW bulls.

The pH measurements of the carcasses collected at 1 and 24 h were not affected by the weaning age of the calves, averaging 6.9 and 5.6 for 1 and 24 h, respectively (Table 3). Likewise, the carcass temperatures were not influenced by the weaning age of the calves, remaining at an average of 39.5 °C and 6.2 °C after 1 and 24 h post mortem, respectively.

### 3.3. Meat Quality

No treatment × time interaction nor weaning group effects were observed for any beef quality traits, but they were all affected by the ageing period (Table 4).

## 4. Discussion

Early weaning emerges as a promising strategy to optimize the metabolic status of cows, playing a crucial role in improving conception rates during the subsequent reproductive season [21,22]. A parallel study conducted by our research group involving 208 lactating cows (including those from the present study) revealed that cows subjected to early weaning showed an approximately 28% greater pregnancy rate in two fixed-time artificial inseminations [5]. In the Brazilian breeding system, where the reproductive season occurs during a period of adequate forage supply and quality, traditional weaning at 210 days of lactation coincides with a time when cows encounter limitations in forage availability and quality [3]. Therefore, weaning to reduce nutritional requirements associated with lactation would allow for the redistribution of nutrients to maternal tissues, resulting in improved body condition of cows before the period of forage scarcity and low quality [23]. This process not only improves the maternal condition but can also improve the flow of nutrients to the fetus starting from mid-gestation [22,24]. Additionally, by eliminating the nutritional requirements associated with lactation, cows reduce their feed intake during the weaning period, thus preserving forage resources during the extended dry season [25].

Concerning calves, the weaning process typically represents an event that can induce stress, exposing the animals to physiological, physical, and psychological challenges with direct implications for their performance and health [26]. Although early weaning is an interesting strategy to enhance the reproductive performance of cows, this practice can entail significant disadvantages for calves, mainly when adequate nutritional conditions are not provided after this period [25]. Studies on early weaning in beef calves have revealed that the implementation of post-weaning supplementation strategies can equalize the performance of traditionally weaned calves, especially when the latter do not receive supplementation [16,21,27]. In the present study, the results indicate that calves in the EW group exhibited reduced body weight compared with those in the CW group from 240 days of age and continuing throughout the experiment. This result was expected and supports our hypothesis, as although supplementation strategies were implemented before the early weaning of calves until the beginning of the feedlot, the CW group also received the same supplemental plane. Therefore, the differences in body weight that persisted throughout subsequent production phases resulted from the additional milk intake period in CW calves compared to the EW group. Furthermore, the difference in response to early weaning, especially regarding the BW of calves in subsequent stages in this study and the results observed in other analyzed research, may be attributed to distinct feeding practices [25,28,29]. While the literature emphasizes advantages in weight gain for early weaned calves due to the immediate implementation of high-concentrate diets, the results of the current experiment challenge this trend. In this study, all calves received similar levels of protein and energy supplementation from day 150 to 240, regardless of the treatment. Therefore, the additional period of milk intake in CW calves may represent an advantage in terms of body weight during this interval.

Despite the reduced BW observed in EW calves compared with CW calves, it is relevant to emphasize that early weaned calves have the potential to achieve productivity levels similar to their conventionally weaned counterparts. Given the expectation that calves experiencing a period of slow growth due to nutritional stress caused by early weaning may regain body weight once nutritional conditions are restored, the compensatory gain could have been anticipated [12,30,31]. However, the compensatory gain was not evident in the present study, as the weight difference caused by the weaning treatment persisted in subsequent productive phases. Nevertheless, this study also demonstrates that, following the immediate weaning period (between 150 and 240 days), early weaned calves can perform comparably to contemporaneously conventionally weaned animals throughout the growing and finishing phases (Figure 2 and Table 2). This outcome can be attributed to the reduction in stress from abrupt weaning, facilitated by the adoption of feeding strategies such as creep-feeding [32,33].

During the finishing phase, although there was a tendency for CW bulls to have a greater feed intake, the age of weaning did not impact FE in the feedlot. Previous studies involving early weaning of beef cattle between 100 and 152 days of age showed higher FE, where weaned calves entered the finishing phase on a high-energy diet at younger ages and lower weights than their conventionally weaned counterparts [15,28,34]. However, in studies where early weaned calves were placed in the feedlot at the same age as their contemporaries subjected to conventional weaning, no FE or conversion differences were observed during the finishing phase [12,29]. These findings align with the results of our research, suggesting that while early weaning may influence FE in the initial stages, such differences cannot persist in the finishing phase. This underscores the importance of considering the specific context of the production system and the developmental stage of the animals for a comprehensive understanding of the effects of early weaning. Therefore, considering factors such as pre-weaning management, the nutritional regime immediately after weaning, and variations in age at the finishing phase in the feedlot, different responses may arise between previous studies and the observations of the current research.

Analysis of the effects of early weaning on carcass characteristics in Nellore beef cattle is limited in the literature. The present study demonstrates that EW reduced carcass weight due to the persistence of the weight difference among the calves in both treatment protocols. This observation contrasts with previous studies that found no differences in hot carcass weight between early weaned and conventionally weaned calves [12,15,29,35,36]. It is essential to highlight that, as indicated in the studies mentioned, early weaned calves (between 70 and 152 days) were fed to high-concentrate supplementation regimes, and in some of these studies, the calves were immediately placed into a feedlot right after weaning. Therefore, the longer feeding duration with high-concentrate diets before slaughter could have been a determining factor for early weaned calves to achieve carcass weights like those of conventionally weaned calves. However, Barker-Neef, Buskirk, Blackt, Doumit and Rust [28] reported that calves weaned early at 100 days had lighter carcasses than those conventionally weaned at 200 days of age, even when slaughtered under similar fat thickness criteria. 

It is suggested that implementing a high-energy diet during the finishing phase in the feedlot, especially for lighter animals that have not reached full muscle development, may lead to an earlier onset of fat deposition, resulting in lower muscularity [37]. This observation is supported by the greater RFTu in early weaned calves (Figure 3). Additionally, based on net energy maintenance calculations, Barker-Neef, Buskirk, Blackt, Doumit and Rust [28] propose that even with identical net energy requirements for gain between early weaned and conventionally weaned animals, conventionally weaned steers require 20% more energy for maintenance simply because of the body weight differences. Therefore, although one of the slaughter criteria in this experiment was maintaining a similar SFT in animals from both treatments, a greater accumulation of RFTu was observed in early weaned bulls. Fat deposition in this region typically occurs earlier than in the dorsal area [38], and it is plausible that, due to lower maintenance demands in early weaned calves, as explained earlier by Barker-Neef, Buskirk, Blackt, Doumit and Rust [28], our findings regarding fat accumulation in this region are validated.

The ribeye area, correlated with muscle tissue deposition in the carcass [39], showed a significant increase detected via ultrasound in the carcasses of animals managed with a conventional weaning strategy in the present study during the feedlot period. This variation is strongly associated with carcass weights [40], influenced by maintaining body weight differences in the calves during weaning. Nevertheless, this difference in the ribeye area at the end of the finishing period was less pronounced and disappeared after slaughter among the tested treatments. Although there was a tendency for CW bulls to have greater feed intake due to their greater BW at the beginning of the feedlot, it is likely that this additional feed intake, as the cattle approach a mature BW, does not meet the needs for protein energy retention. Therefore, the differences observed in the weights of the primal cuts, being greater in CW bulls, can be related to the persistence of BW differences established between the periods of early and conventional weaning rather than an increase in the rate of muscle deposition in the carcass during the feedlot period (Table 3). This finding is reinforced by the absence of a difference in the yield percentage between animals weaned early or conventionally, considering the correlation of this measure with the amount of muscle in the carcass [41].

The meat quality traits, such as color, tenderness, and cooking losses, are inherently associated with the rate of pH decline and the final pH [42,43,44]. Post-mortem alterations in the rate of pH decline and the final pH can impact biochemical activity and muscle structure, consequently affecting these meat quality properties [43]. Furthermore, pre-slaughter management and nutrition are factors with a significant influence on carcass pH. In this study, we did not observe significant impacts on these characteristics concerning the weaning time of the calves. This outcome can be attributed to the uniformity in both the post-weaning feeding regime and the pre-slaughter management applied to the bulls, meaning that the final pH values of the carcasses in both treatments remained within the normal limits (pH = 5.6) after the muscle conversion into meat [43]. The lack of effect of the calves’ weaning time on meat tenderness aligns with other studies, indicating that the final pH and, consequently, meat tenderness were similar between early weaned and traditionally weaned animals [28,45,46,47,48]. 

Another factor contributing to the absence of differences in meat quality traits is related to the subcutaneous fat deposition on the carcasses. This study indicates that one of the criteria for slaughtering the bulls was a similar level of subcutaneous fat in both treatments. Subcutaneous fat acts as thermal insulation during carcass chilling, allowing for a more gradual temperature drop. This reduces the incidence of cold shortening of muscle fibers, strongly associated with tenderness, color, and water retention in the meat [49].

## 5. Conclusions

The early weaning practice affects the body weight of calves and results in lower BW, but does not negatively affect ADG or FE. Despite the lower carcass weight due to lower BW in calves weaned early, characteristics such as dressing percentage, rib eye area, subcutaneous fat thickness, and meat quality are not compromised. Given the challenges posed by the lower BW, the importance of future research focused on nutritional and health management strategies that can offset this impact is highlighted. Exploring adjustments in diet or specific supplementation for early weaned calves may be crucial to optimize productive outcomes. 

## Figures and Tables

**Figure 1 animals-14-00779-f001:**
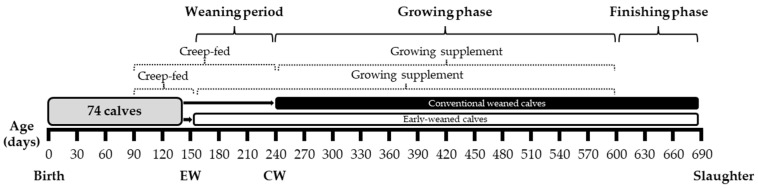
Experimental design of Nellore calves subjected to early weaning (EW; 150 days) or conventional weaning (CW; 240 days).

**Figure 2 animals-14-00779-f002:**
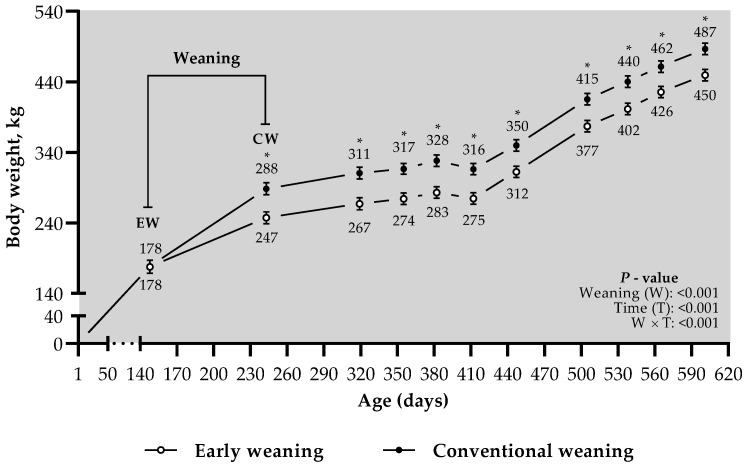
Body weight during the growing phase of calves subjected to early weaning (150 days) or conventional weaning (240 days). * Treatment means differ by student t test (*p* < 0.001).

**Figure 3 animals-14-00779-f003:**
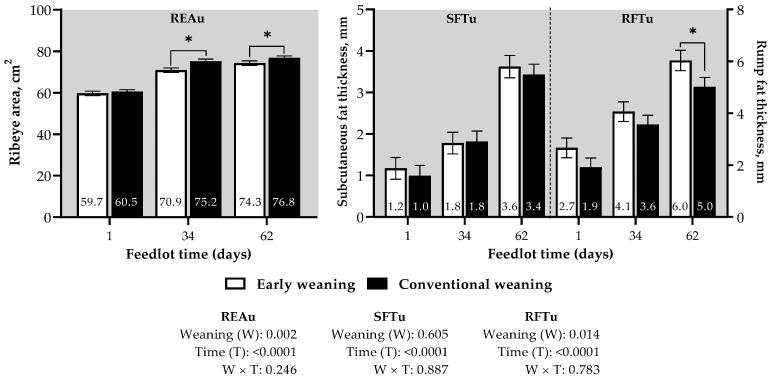
Ultrasound measurements during the finishing phase of calves subjected to early weaning (150 days) or conventional weaning (240 days). * Treatment means differ by student t test (*p* < 0.001).

**Table 1 animals-14-00779-t001:** Supplement ingredients for cow-calf and growing phases, and adaptation and finishing diets in feedlot management.

Item	Supplements		
	Cow-Calf	Growing	Finishing Diet
Ingredients, % DM			
Corn silage	-	-	29.6
Ground corn	59.5	64.9	64.3
Soybean meal	35.0	28.5	2.6
Urea	-	2.5	1.4
Mineral mix ^1^	1.5	2.1	0.6
Salt	4.0	2.0	0.5
Calcium limestone	-	-	0.6
Potassium chloride	-	-	0.4
Chemical composition, % DM			
Dry matter, % NM	86.18	82.79	65.02
Crude protein	23.76	32.81	15.32
Neutral detergent fiber	22.49	15.86	35.30
Acid detergent fiber	10.11	10.87	15.78
Ether extract	2.48	2.46	2.78
Total digestible nutrients	80.71	80.70	77.34

^1^ The composition of the mineral mix contained (per kg): calcium (max/min), 230–200 g; phosphorus, 160 g; sulfur, 60 g; zinc, 8100 mg; sodium monensin, 4000 mg; copper, 2700 mg; manganese, 2700 mg; fluorine (max), 1600 mg; cobalt, 160 mg; iodine, 135 mg; selenium, 80 mg.

**Table 2 animals-14-00779-t002:** Performance during feedlot finishing of young bulls undergoing early (150 days) or conventional (240 days) weaning.

Item	Weaning		SEM	*p*-Value
	Early	Conventional		
Initial body weight, kg	440	476	18.9	<0.001
Final body weight, kg	570	611	198	<0.001
Average daily gain, kg/day	1.55	1.60	0.15	0.284
Dry matter intake, kg/day	10.8	11.2	0.33	0.083
Feed efficiency, g ADG/kg DMI	144.3	143.3	9.79	0.803

**Table 3 animals-14-00779-t003:** Characteristics and yields of carcass and hindquarter cuts of young bulls undergoing early weaning (150 days) or conventional weaning (240 days).

Item	Weaning		SEM	*p*-Value
	Early	Conventional		
Carcass characteristics				
Hot carcass weight, kg	323	347	10.56	<0.001
Dressing percentage, %	56.7	57.0	0.46	0.447
Ribeye area, cm^2^	83.2	82.5	1.33	0.686
Subcutaneous fat thickness, mm	4.0	4.0	0.33	0.999
Renal, pelvic, and inguinal fat, kg	12.1	11.4	0.39	0.233
Renal, pelvic, and inguinal fat, % ^1^	3.7	3.2	0.12	0.008
pH and temperature				
pH_1h_	6.9	6.9	0.02	0.489
pH_24h_	5.6	5.6	0.03	0.586
Temperature_1h_	39.6	39.3	0.38	0.202
Temperature_24h_	6.34	6.21	0.52	0.359
Primal cuts weight, kg				
Forequarter	60.8	65.7	1.27	<0.001
Thin Flank	22.2	23.6	0.95	<0.001
Hindquarter	74.3	79.0	1.34	<0.001
Primal cuts yield, % ^2^				
Forequarter	38.3	38.6	0.34	0.284
Thin Flank	14.4	14.3	0.50	0.533
Hindquarter	47.4	47.1	0.30	0.410
Hindquarter tissues, kg				
Retail cuts	51.8	55.3	1.05	<0.001
Bones	13.8	14.7	0.46	<0.001
Trimmings	5.2	5.1	0.22	0.763
Hindquarter tissues ^3^				
Retail cuts	68.8	69.0	0.36	0.654
Bones	18.6	18.7	0.24	0.724
Trimming	6.9	6.4	0.21	0.055

^1^ HCW: Percentage of renal, pelvic, and inguinal fat in relation to carcass weight; ^2^ Percentage of primal cuts in relation to the weight of the left half carcass; ^3^ Percentage of tissues in relation to hindquarter weight.

**Table 4 animals-14-00779-t004:** Meat quality at different post-mortem days of young bulls undergoing early weaning (150 days) or conventional (240 days).

Trait	Weaning		SEM	Ageing Days			SEM	*p*-Value		
	Early	Conventional		1	7	14		Weaning	Ageing	Weaning × Ageing
Color L*	44.6	44.1	0.53	42.1	46.1	44.9	0.55	0.153	<0.001	0.416
Color a*	22.6	23.1	0.55	21.8	22.6	24.2	0.57	0.103	<0.001	0.592
Color b*	16.6	16.8	0.33	14.5	17.1	18.0	0.35	0.427	<0.001	0.091
Cooking losses, %	26.2	27.4	2.91	24.8	25.8	29.9	2.95	0.150	<0.001	0.104
WBSF ^1^, N	58.1	60.6	5.00	67.1	56.6	54.4	5.10	0.144	<0.001	0.915

^1^ WBSF: Warner–Bratzler shear force.

## Data Availability

All relevant data are presented within the paper.

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
