# Peer review of "Effects of Early Weaning on Performance and Carcass Quality of Nellore Young Bulls"

_animals, 2024, doi:10.3390/ani14050779_

Round 1

Reviewer 1 Report

Comments and Suggestions for Authors

This paper deals with early weaning of bull calves in relation to growth performance and meat quality compared to conventional weaning age of bull calves. This manuscript is well written and every section is relevant.

However, stricktly speaking the fact is that EW calves did not give a similar final body weight/carcass weight and necessitated more trim than CW calves. Ok, they had similar ADG, FE and meat characteristics, but the final weight/carcass weight was lower. Therefore, the conclusion is over-optimistic according to results. 

The authors introduced the impact of EW on dams reproductive performance. However, on one hand this was not the objective of this study,. On the other hand, the stress caused to cows and calves that underwent EW is not reported, neither that the weaning method used. THerefore, impact on dams' reproductive performance should not be included in this manuscript. This as such would be matter to another publication.

L 138-146 in the finishing phase, bulls were fed adlibitum. But it is written that they were weighed before feeding. THis should not have any effect since they were fed adlibitum. Therefore why specify this condition? Or were the bulls not really fed ad libitum? Similarly fasting for 14 hours may cause the bulls to eat too much on refeeding causing rumen acidosis. Why measuring weight sometimes after fasting, sometimes not? The weight could have been taken always full if they were really fed ad libitum, they would then be always full BW without affecting rumen environment.

Comments on the Quality of English Language

english language is fine in this manuscript

Author Response

Plase find attached our answers about reviewer 1 comments.

Reviewer 2 Report

Comments and Suggestions for Authors

Dear Author,

This is an interesting paper, and has some good findings relative to the effect of weaning age. However there are a number of defects that you will need to address. I have made comments on the PDF for your consideration.

Comments on the Quality of English Language

Minor editing of English language required

Author Response

Plase find attached our answers about reviewer 2 comments.

Reviewer 3 Report

Comments and Suggestions for Authors

In my opinion, in general, this study is of a good potential but overall conclusion actually does not reveal anything novel. The study is well described, M&M are appropriately chosen and described. But, the hypothesis, experiment design and conclusion are questionable. Nowadays, the weaning strategies for beef cattle assume even shorter weaning time – of course with adjusted nutritional management. And, we know that earlier weaning of calves reduces the nutrient requirements of the cow, improves cows’ condition and shortens the preparation for the next calving. That’s why, in my opinion, this study lacks practical implication as there is no benefits of slaughtering animals at lower body mass or at similar body weight but achieved in longer rearing period. There is a lack of nutritional manipulation for the EW group at the weaning stage to compensate the efficiency of breeding compared to the CW group.

The main conclusion of this study is that early-weaned calves had lower gains than CW calves, but similar meat quality. Thus, the assumed approach in this study was restricted feeding throughout whole experimental period (citing the Authors: Lines 310-315:„In the present study, calves subjected to EW and those subjected to CW began protein-energy supplementation at 90 days of age, receiving 5g of supplement per kg of body weight, maintaining this supplementation regimen postweaning”; Lines: 320-323: „In this study, all calves received similar levels of protein and energy supplementation from day 150 to 240, regardless of the treatment. Therefore, the additional period of milk intake in CW calves may represent an advantage in terms of body weight during this interval.”) which places the EW group in a worse position from the very beginning of the study.

The Authors state also: „ Despite the reduced BW observed in EW calves compared with CW calves, it is relevant to emphasize that early weaned calves have the potential to achieve productivity levels similar to their conventionally weaned counterparts.” – exactly, if they would have been provided with adequate amounts of nutrients (as in study by Pedro et. all, 2023), or offered the diet ad libitum probably yes, but not in this design of the study.

Conclusion from the study is too general. It should be point more precisely the results, emphasizing overall fattening period, pros and cons of both strategies.

Technical notes:

Simple summary:

Lines 23-27 Doubled sentence

Abstract:

Lines 44-45: Therefore, when combined with efficient supplementation to promote growth and productivity, early weaning emerges as a sustainable and beneficial practice in beef cattle production. – This sentence seems too far-fetched

Key words:

Remove hindquarter, suckling cow; Instead please use meat quality or carcass characteristics, beef cattle

Introduction:

Line 50: „over ~80%” replace ~with „over ca. 80%” phrase

M&M:

Animals for slaughtering were chosen (?) by the researchers based on BW and SFTu criteria (“After an approximately 87-days of feeding, bulls were selected for slaughter based on the BW criteria and ultrasound subcutaneous fat thickness (SFTu) above 3 mm…) – please indicate precisely how many animals were subjected to carcass analysis as it might “hidden” the possible effects of two rearing approaches.

Results:

Table 1

Minera mix1 – check spelling

Table 2

Please add the information regarding the overall rearing period and time (days) till slaughter

Comments on the Quality of English Language

Manuscript is well written. 

Author Response

Plase find attached our answers about reviewer 3 comments.

Reviewer 4 Report

Comments and Suggestions for Authors

The article is well written and has clear objectives. However, the conclusions are somewhat biased. While during the fattening period, it is during the inter-weaning period that the differences are found. Furthermore, the weight loss suffered by the EW flock is never regained. This should be put in the conclusions, as it has economic repercussions in terms of being able to sell less meat.

L25-27 repeated sentence. Please delete.

L42 Which carcass characteristics?

Use calves instead bulls, please.

L169 dressing percentage is preferable (as used in Table 3) to carcass yield, which often is confounded with the saleable meat from a carcass.

Author Response

Plase find attached our answers about reviewer 4 comments.

Round 2

Reviewer 3 Report

Comments and Suggestions for Authors

Dear Authors,

I understand your perspective. The design of the study might be further discussed but it will not change anything at this time.

I find the revised version suitable for publication.

Kind regards